# Development of an Inner Finishing Method for Brass Cone Pipe via a Movable Manual Electromagnet in a Magnetic Abrasive Finishing Process

Jeong Su Kim [1], Sieb Chanchamnan [1], Lida Heng [2], Guenil Kim [3], Sung-Hoon Oh [3] and Sang Don Mun [1,2,*]

1   Department of Energy Storage, Conversion Engineering of Graduate School, Jeonbuk National University, Jeonju 54896, Korea; kjs1592@jbnu.ac.kr (J.S.K.); chanchamnansieb@gmail.com (S.C.)
2   Division of Mechanical Design Engineering, Jeonbuk National University, Jeonju 54896, Korea; henglida1@gmail.com
3   Department of Mechanical System Engineering, Jeonbuk National University, Jeonju 54896, Korea; qikim33@gmail.com (G.K.); oshun0305@jbnu.ac.kr (S.-H.O.)
*   Correspondence: msd@jbnu.ac.kr; Tel.: +82-63-270-4762

**Abstract:** This paper describes the development of a movable manual electromagnet with an adjustable flux density to improve the inner surface smoothness of a cone pipe using a magnetic abrasive finishing process. This method is fabricated to reduce further the roughness of the internal surface of the conic shape, which was modeled as an electromagnet oscillating in the work zone with a ball roller. Statistically significant improvement in the process was achieved using unbounded magnetic abrasive, light oil, flux density, controlled feed rate, and constant rotational speed in the experiment. The ball transfer equipped on the top of the electromagnet pole plays an essential role in spinning over the outer cone pipe during the experiment and helps reduce friction while the workpiece fluctuates. Furthermore, the flux density can be changed to control the magnetic force and select the most acceptable option. In addition, a procedure for finishing has been designed for finishing a cone pipe, and we sought to understand how the flux density affects the material in removal exterior roughness. As a result, the flux density is clarified, and a higher flux density achieves excellent removal of surface roughness of the inner deformed pipe from 1.68 μm to 0.39 μm within 24 min.

**Keywords:** magnetic abrasive finishing; flux density; movable manual electromagnetic; cone pipe; surface roughness; material removal weight; ball transfer

## 1. Introduction

Surface quality is a crucial factor in mechanical components for enhancing disrupted flow on both the external and internal workpieces to meet specular surface requirements to deliver efficient flow in a tube [1,2]. Complex internal surfaces with a curve feature are widely used in medical devices, aerospace, automotive industries, and hydrogen gas flow applications [3,4]. Tubes convergence and divergence are the most suitable for changing the velocity in systems that need a mirrorlike surface. In addition, an exhaust cone is desirable for separating the connected piping without having to cut, unsolder or otherwise damage the materials; this is especially important concerning the smoothness of the inner workpiece [5]. To further achieve surface smoothness of an interior surface, the magnetic abrasive finishing (MAF) process has been used to get a better finishing surface for decades and provides a good result for a textural surface [6,7]. Because it involves small magnetic particles that are highly elastic and closely follow the surface, the method can be used for flat surface finishing, complex curved surfaces, and the inner surface of a tube [8,9]. However, conventional methods involve polishing, which is too hard for many geometries and materials such as brass, wire glass, ceramics, and stainless steel [10,11].

Moreover, polishing with a conventional operation makes it difficult to obtain an accurate surface and shape, as polishing may cause small-scale splits and burrs on the workpiece [12]. In particular, brass is a soft, very ductile material that is not as easy to polish as more rigid materials [13,14]. To overcome this issue, a non-conventional method is required to achieve the perfect finishing of the workpiece.

Many researchers have evolved many advanced finishing techniques to solve complex and straightforward geometrical bodies. Manas and Jain [15] have developed rotational-magnetorheological abrasive flow finishing (R-MRAFF) process for stainless steel and brass flat. The results obtained that the high rotational speed of the magnet and its square term have significantly improved in surface roughness. In recent research, Iskander et al. [8] showed that the concepts of magnetic concentration and electro-permanent magnets could polish free-form surfaces with hard-to-access geometries using magnetic fluids. This system was designed to investigate the removal of paint from a cylindrical surface, which used an H-bridge circuit to toggle magnetic field strength and a conventional fan-based cooling system. The paint on the outer surface of the cylinder was removed as the system was worked without moving and rotating. Rajendra et al. [16] analyzed the abrasive flow machine using a neural network to predict the result and compare it with the experimental work. As a result, the simulation is similar and can be computed as a constraint to advancing machining efficiency. Junmo et al. [4] developed the high-speed internal finishing of the capillary tube using MAF to clarify the high-speed, and the result was successfully achieved by using a double pole-tip with the rotational speed of 10,000 RPM and is twelve times more capable than the multiple pole-tip finishing system. Yan et al. [9] focused on the roughness of an inner surface treated using MAF and examined the effects of various parameters on the material removal rate. The system was designed for finishing three kinds of tubes (Ly12 aluminum alloy, 316L stainless steel, and H62 brass). Various input parameters were studied, including polishing speed, magnetic abrasive material, and grain size. The results showed that the magnetic material, grain size, and rotational speed affected the inner surface finish. Jain et al. [17] investigated the MAF process on a non-magnetic stainless-steel object and discovered that the working distance and rotational speed significantly impact the surface roughness values. Changes in surface roughness happen as the current to the electromagnet increases and the working gap decreases. The surface finishing of plane surfaces may improve greatly with an increase in grain, size, feed rate, and various parameters such as working gap, machining time and grit size significantly affect the results.

However, the MAF process is still considered insufficient to obtain a high-quality surface finish, especially in finishing complex internal surfaces with curve features, as mentioned above. Each process has its limitations with different technologies. In the conventional MAF process using finishing tools, it is difficult to obtain excellent surface smoothness. Brass cone pipes are used in many applications that demand a very smooth surface. According to many researchers, the inner surface roughness of pipe was improved by MAF, but conical shapes are challenged. This research developed a new method of polishing the inner complex surface using a magnetic abrasive finishing process via a moving electromagnet to overcome these problems.

This paper introduced the development of electromagnet, material preparation, experimental setup, and flux density is clarified in the result section. This research focused on a movable manual electromagnet, which can adjust the flux density to the electromagnet field based on magnet force changes. In addition, a manual electromagnet is considered to be more flexible due to the different flux densities applied to the work zone during processing; especially, it was designed to oscillate on the *Y*-axis and expend over the fluctuating volume of the workpiece. The ball transfer was modeled at the electromagnet's core, which spins and fluctuates over the periphery. The fluctuating flexibility of the ball transfer both reduces friction over the surface and achieves circulation to ensure a balanced rotation. In addition, following this method helps enhance the surface finishing process compared to the permanent electromagnet. The flux density is investigated and clarified

in electromagnetism in the precision inner surface roughness of the deformed pipe. The primary parameters that need to be optimized to obtain increased smoothness include unbonded abrasive, rotational speed, feed rate, flux density (changeable), and a processing time of 24 min. Then, the feasibility of the method evaluated the material removal weight, surface quality, and percentage of how-well finishing of the experiment. In conclusion, it was gathered rationale constraint for improving surface texture.

## 2. Development and Experimental Setup of a Movable Electromagnet

### 2.1. Development of Electromagnet

Electromagnets were chosen for this study, unlike the commonly used permanent magnets, which have a constant flux density. Using rare earth permanent magnets has advantages, such as the lack of power supply, low cost, and long lifetime. However, there are many disadvantages, such as being uncontrollable, having a non-adjustable flux density, and magnetic particles are attracted to the poles [18]. In addition, in an earlier system, the magnetic system could not move in the polishing zone, whereas electromagnets can be moved both forward and backward, and the balancing diameter increased. Moreover, an electromagnetic placed on the slider could work over an extended scale.

Figure 1 shows the development of the electromagnet and ball roller are modeled and assembled. The manual electromagnet was assembled using an iron pipe with an outer diameter of 30 mm and an inner diameter of 20 mm with a length of 30 mm. It was mated to a solid cylinder with a diameter of 20 mm. Holes were drilled in the cylinder to accommodate iron balls 6 mm in diameter, and the balls were installed into the hole to rotate 360 degrees without disappearing. The iron balls play an essential role in reducing friction surrounding an exterior workpiece. In addition, both electromagnetics was on the sides of the deformed pipe to generate the force between unbonded magnetic abrasive particles in the precision zone. The workpiece is a complex body with narrow and vast parts; this design was crucial to oscillate on a perimeter pipe. The double pole-tip side by the side of the cone pipe increased the flux density due to the changeable voltage from the controller is attracted magnetic force dramatically to the pole edge [4].

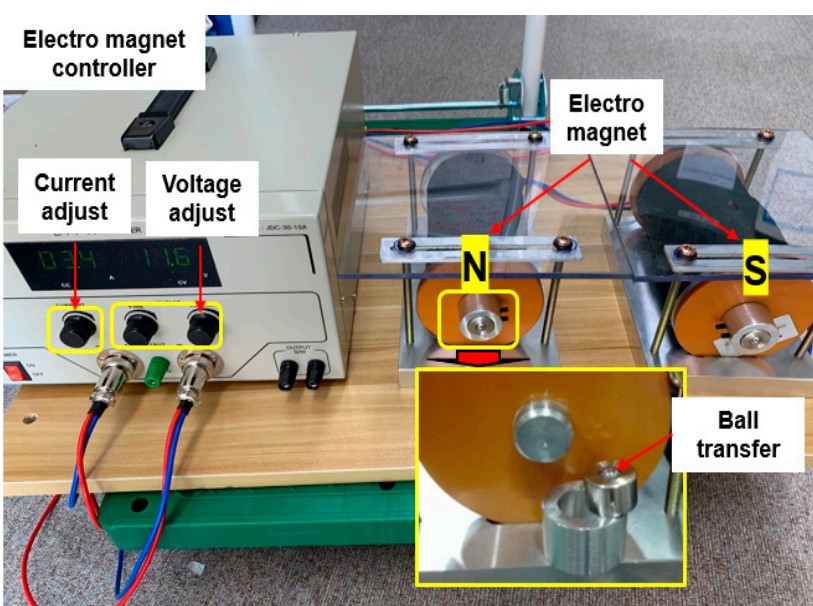

**Figure 1.** Modeling manual electromagnet with ball roller.

During the finishing process, the magnetic abrasive tools were used for polishing at neither the S pole nor the N pole to make sure the process was even. Moreover, adjusted voltages (5 V, 8 V, 12 V, and 16 V) reflected fields of 110 mT, 185 mT, 260 mT, and 330 mT, respectively. The result of magnetic flux density was measured using a tesla meter in the

magnetic controller (model: JDC-30-10A, JL magnet manufacturer, Seoul, South Korea). Then, in the procedure to determine the variations in surface roughness after the finishing process, the surface roughness parameter Ra (average surface roughness) was measured every 6 min for a total of 24 min using a surface roughness tester (model: Mitutoyo SJ-400 by Mitutoyo Sakado, Kawasaki, Japan). Changes in the removed weight (Rw) were measured every 6 min for a total of 24 min using a scale (ADD GH-252).

### 2.2. Experimental Setup

The experimental setup in the current work is shown in Figure 2. The system developed the precision magnetic abrasive finishing process to achieve a high surface smoothness in this study. Each component in the process was assembled step by step, including the spindle machine, manual electromagnet, magnet controller, stepper motor, sliders, pulley, board controller, sensors, ball transfer mechanism, magnetic abrasive, and brass cone pipe. The primary mechanism was divided into a spindle machine, a magnetic motion system, a manual electromagnet, and an abrasive composite. First, the spindle machine (model: EMT-D210V, range 5 RPM–2500 RPM, WMT CNC Industrial Co., Ltd, Chizhou, China) was installed in the supporting flat frame. Then, the magnetic motion system was controlled using an electric controller, which had a stepper motor to control the penetration of the magnet. Moreover, the stepper motor was connected by a pulley to the gears, where it held a screw used for rotating the slider backward and forward with the two recognition sensors. A ball transfer mechanism was included in the core of the manual electromagnet, and it plays an essential role in spinning over the workpiece surface as the magnetic controller (JDC-30-10A) adjusted the voltage to the electromagnet. To that end, the composite of abrasives was mixed by hand in a petri dish. It was placed into the deformed pipe, which was fixed to the spindle.

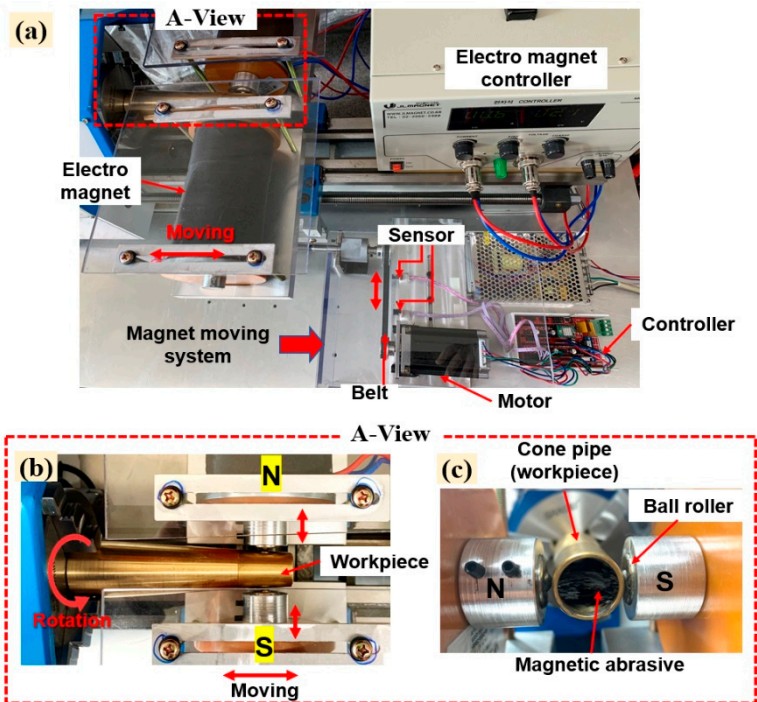

**Figure 2.** Photo view of a magnetic abrasive finishing process using moving electromagnets for the deformed pipe: (**a**) full view, (**b**) top view, and (**c**) front view.

### 2.3. Material Preparation and Experimental Conditions

The workpiece made from brass $CuZn_3O$ and its mechanical properties and chemical composition are summarized in Tables 1 and 2, respectively. The parameters relevant to the precision finishing process include the cone pipe dimensions, flux density, magnetic

abrasive, rotational speed, feed rate, and machine time, which are described in Table 3. The cone pipe was selected as the workpiece with a top diameter of 21.5 mm and a bottom diameter of 24.5 mm. The inner diameter of the top was 16 mm, and the bottom diameter was 22 mm. The initial surface roughness Ra before finishing was 1.68 μm. Iron powder (Fe#200 μm, 2.5 g), magnetic abrasive (Fe/Al$_2$O$_3$, 4.5 g), and light oil (1 mL) were selected as an unbounded magnetic abrasive. Moreover, the rotational speed was fixed at 800 RPM, and the feed rate was 1 mm per second. The total machining time was 24 min, which was measured every 6 min. The electromagnet controller (model: JDC-30-10A) was used for adjusting voltage to control flux density for every experiment, starting from DC-5V (110 mT), DC-8V (185 mT), DC-12V (260 mT), and DC-16V (330 mT), respectively.

**Table 1.** Prescribed mechanical properties of brass cone pipe (CuZn$_3$O).

| Tensile Strength (MPa) | Yield Strength (MPa) | Elongation A$_{50}$ (%) | Hardness (HV) | Young's Modulus (GPa) |
|---|---|---|---|---|
| 350 | 160 | 40 | 55–90 | 110 |

**Table 2.** Chemical composition of brass cone pipe (CuZn$_3$O).

| Copper (Cu) | Iron (Fe) | Lead (Pb) | Zinc (Zn) | Al | Nickel (Ni) | Tin (Sn) |
|---|---|---|---|---|---|---|
| 69%–71% | 0.05% max | 0.05% max | rest% | 0.02% max | 0.3% max | 0.1% max |

**Table 3.** Experiment Conditions.

| | |
|---|---|
| Workpiece | Brass cone pipe dimensions: ($D_{top,outer} = 21.5$ mm; $D_{top,inner} = 18$ mm; $d_{bottom,oute} = 24.5$ mm; $d_{bottom,inner} = 22$ mm; length $= 30$ mm) |
| Flux density | 110 mT (DC-5V), 185 mT (DC-8V), 260 mT (DC-12V), 330 mT (DC-16V) |
| Magnetic abrasive | Iron powder (Fe#200 μm): 2.5-g Magnetic abrasive (Fe/Al$_2$O$_3$): 4.5-g, Light oil: 1-mL |
| Rotational speed | 800 RPM |
| Machining time | 6, 12, 18, 24 min |
| Feed rate | Feed: 16 mm, Feed rate: 1 mm/s |

An analytical balance (SHIMADZU-AUW220D), surface roughness tester (Mitutoyo SJ-400), and a scanning electron microscope (SEM at 100×, and 500×) were prepared to analyze each parameter of the workpiece during the experiment. Figure 3 shows the brass cone pipe modeling and actual part. Figure 4 shows an instrument measuring surface roughness Ra for brass cone pipe. As demonstrated in the photo, the probe set was moved inside pipe 5 mm with a moving acceleration of 0.5 m/s to analyze the surface roughness of the sample. Five different cone pipes were finished for the experiment and followed by the parameter input in Table 3 below.

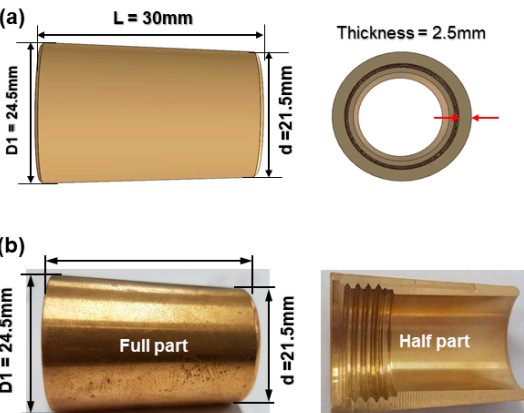

**Figure 3.** Brass cone pipe (**a**) modeling, (**b**) whole part and half part.

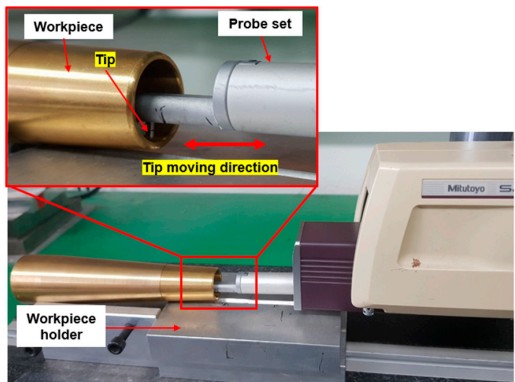

**Figure 4.** Instrument testing surface roughness of brass cone pipe (Mitutoyo SJ-400).

## 3. Results and Discussion

### 3.1. Effect of Changes in Magnetic Field Flux Density on the Finishing Process

Figure 5 shows the surface roughness and the processing time for polishing the inner surface of the cone pipe up to 24 min. The rotational speed of 800 RPM was fixed for the whole experiment. Overall, the 110 mT flux density resulted in a negligible decrease in roughness from 1.68 μm to 1.64 μm. The 185 mT flux density was nearly the same as the 110 mT while polishing for 6 min. However, it declined rapidly for the remaining 18 min, resulting in the roughness of 1.14 μm. Data for DC-12V (260 mT flux density) decreased consistently to a finished roughness of 1 μm.

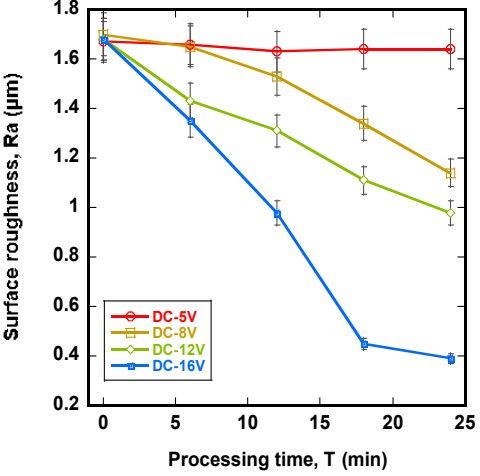

**Figure 5.** Comparing surface roughness by changeable flux density with a rotational speed of 800 RPM.

Last but not least, the 330 mT flux density showed a significant and continuous drop from 1.68 μm to 0.39 μm. The error bars are shown (±5%) of all measurements in the experiment. Thus, a higher flux density results in a highly polished inner surface of a deformed pipe; this can be explained due to the electromagnetism between two magnetic poles, which charges the abrasive particles touching the surface, providing friction force resulted in removal roughness. An electromagnet and the powder grains from the flux density during their movements relative to magnetic poles. In summary, the final surface roughness at 110 mT, 185 mT, 260 mT, and 330 mT were 1.63 μm, 1.14 μm, 0.98 μm, and 0.39 μm, respectively. In conclusion, a high flux density is required for excellent surface finishing.

We then tried to find a general relationship between the percentages of finishing relative to the initial surface roughness and the final surface roughness. POP stands for polishing percentage, ISR is for initial surface roughness, and FSR stands for final surface roughness. POP indicates how successfully each condition polishes the workpiece. Before and after the experiment, ISR and FSR were calibrated, with FSR being measured after 24 min of processing:

$$\text{POP} = \frac{\text{ISR} - \text{FSR}}{\text{ISR}} \times 100\%$$

Figure 6 shows the percentage of the completed process at various voltages. According to the graph, there were patterns in the percentage of cases that were removed. DC 5, 8, and 12 voltages climbed gradually by 2.38%, 32.14%, and 41.67% of the finished value, respectively. The DC 16 voltage shot up to 76.79%. Overall, the proportion of all grew, although the highest voltage climbed at a faster rate. If the procedure works, we can assume that a high voltage is an optimum option.

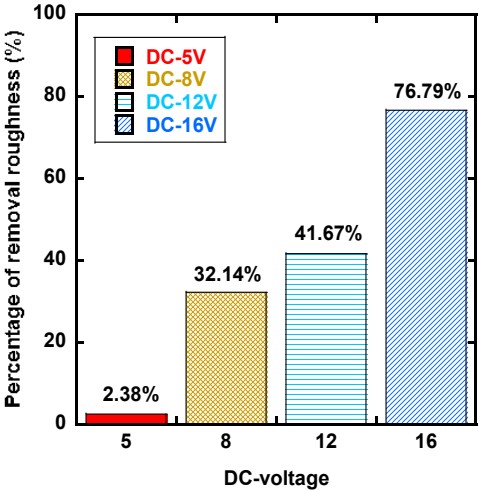

**Figure 6.** Percentage improvement in surface roughness according to different voltages.

### 3.2. Material Removal Weight vs. Processing Time

Figure 7 shows the material removal weight of the cone pipe after 24 min of processing. Determining the best alternative for the internal surface finishing operation: the spinning acceleration of the rotating magnetic field was balanced by a rotational speed of 800 RPM and variable flux density; this was recorded every 6 min for a total of 24 min. Furthermore, the most considerable increase was in the 330 MT flux density, where the inner diameter reduced weight to 4.22 mg, but the most negligible reduction was 0.95 mg in 6 min in the 110 mT flux density. The 185 mT field, on the other hand, resulted in 2.10 mg removed, while the 280 mT showed a 3.50 mg increase in removal weight in the first measurement. All the modifications were more moderate in the following measurement step, going up by a factor of two. However, after 18 min of testing, data at both 6 and 12 min improved significantly. Finally, the discharge inner surface diameter was the greatest of all, with a remarkable result of 16.89 mg for 330 mT after 24 min, while the other three flux densities

were well behind. Overall, the maximum flux density input to the electromagnet was the optimum option, considerably improving surface smoothness while maintaining adequate processing time. The flux density is proportional to the magnetic force in the presence of an unbonded abrasive and constant rotating velocity.

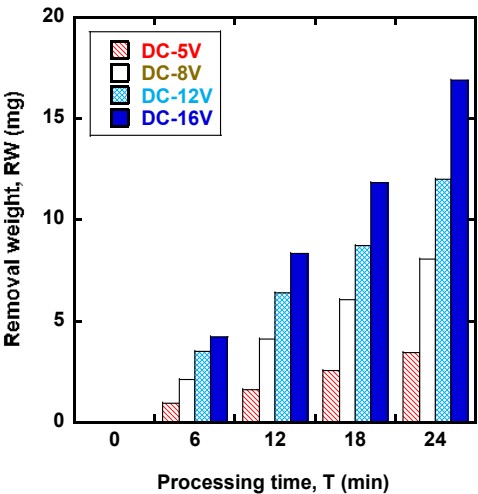

**Figure 7.** Material removal weight vs. Processing time.

The surface smoothness after employing the magnetic abrasive finishing technique is shown in Figure 8. The workpiece was halved and tested using an SEM micro machine. According to the result, this process is feasible to reduce the surface roughness from the initial surface roughness (Ra) of 1.68 μm to 0.39 μm on both the wide and narrow parts. Figure 8a shows that the surface condition was not good before the finishing process, and unevenness was found everywhere on the surface. Figure 8b,c show the final surface conditions of the MAF process within 24 min of finishing time by using an SEM micro machine. After the finishing process, the initial unevenness was removed entirely. Therefore, the surface conditions are smoother and more delicate than the initial surface before finishing (see Figure 8b,c). Figure 9a, the image of the surface condition, shows that the reflection of a red pen before the finishing process was too blurry, and its reflection was not projected well on the surface of a cone pipe workpiece. Figure 9b shows the surface conditions after the MAF process. It can be seen that the inner surface condition of the workpiece was very smooth and clean (like a mirror surface), and the reflection of the red pen can be clearly seen.

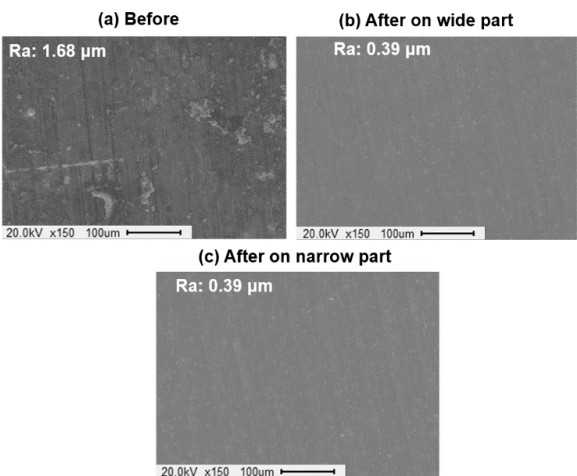

**Figure 8.** SEM micro picture of the inner surface of the cone pipe, (**a**) before finishing, (**b**) on the wide part after finishing, (**c**) on the narrow part after finishing.

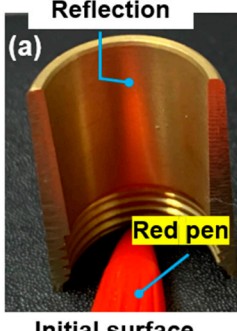
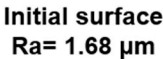
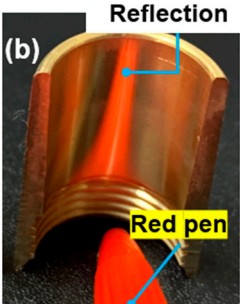

**Figure 9.** (**a**) Mirror image before finishing, (**b**) Mirror image after finishing.

## 4. Conclusions

By testing the finishing of the deformed pipe using the MAF process with various magnetic fields, the system identified the influence of adjustable flux density. Conclusions are summarized as follows:

1.  The inside surface of a brass cone pipe can be finished successfully by a magnetic abrasive finishing technique using a portable electromagnetic system, which involves using a manual electromagnet, abrasive, and a ball roller to create this result. The electromagnet can be oscillated over the periphery and expended with fluctuates volume of the sample.
2.  The optimum option for completing the inner surface of the cone pipe was determined to be high flux density, which reduced starting roughness from 1.68 μm to 0.39 μm in 24 min.
3.  The complex internal surface with a curve feature can be finished, following this method to improve inner surface roughness.
4.  The characteristic of flux density has clarified the phenomena to develop other methods in the subsequent research.
5.  SEM micro-images show that the surface quality of the cone pipe was successfully improved by the MAF technique using a portable electromagnetic system under the optimal conditions (rotational speed: 800 RPM, magnetic flux density: 330 mT, finishing time: 24 min).
6.  The developed model achieved a specular surface, which showed an efficiency of 76.79% for finishing the surface inside a cone pipe.

**Author Contributions:** Conceptualization and writing, J.S.K.; methodology, J.S.K., S.C. and L.H.; software, S.C. and G.K.; validation, S.-H.O.; writing—original draft preparation, J.S.K. and S.D.M.; funding acquisition, L.H. and S.D.M. All authors have read and agreed to the published version of the manuscript.

**Funding:** This research was funded by the National Research Foundation (NRF) of South Korea, (Grant No. 2019R1F1A1061819, and Grant No. 2021R1I1A1A01060699).

**Institutional Review Board Statement:** Not applicable.

**Informed Consent Statement:** Not applicable.

**Data Availability Statement:** Data are available from the authors upon request.

**Conflicts of Interest:** The authors declare no conflict of interest.

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
