# Peer review of "Development of an Inner Finishing Method for Brass Cone Pipe via a Movable Manual Electromagnet in a Magnetic Abrasive Finishing Process"

_metals, doi:10.3390/met11091379_

Round 1

Reviewer 1 Report

Please see the file attached.

Author Response

Dear Reviewer,

We would like to thank the reviewer for peering through this manuscript and commenting on the interesting point for improving the novelty of the manuscript.

The manuscript has been revised according to your comments. We have sent the attachment (Response Reviewer-comments).

Best regards, 

Reviewer 2 Report

In this manuscript, the authors proposed a movable manual electromagnet with an adjustable flux density to improve the inner surface smoothness of a cone pipe with a magnetic abrasive finishing process. The processing device and method, in this paper, has certain guiding significance on finishing the cone pipe. The experimental results show that the device can effectively reduce the surface roughness of the cone pipe. However, there are some major revisions need to be improved before it considered to accept:       1.In Figure 4, the roughness value is the average value calculated statistically from the measurement results? If so, the figure should include error features of the measurement, such as error bars.      2.In Figure 7, there are two subgraphs numbered (a) and two subgraphs numbered (b). How should they be distinguished?      3.This paper looks more like an experimental report, and the authors are advised to further analyze and discuss the experimental results so that the reader can understand the mechanism of MAF process, clearly.      4.There are spelling errors are found in the manuscript, including words and variables. Please check the whole paper carefully.

Author Response

(The authors gave the same response as above.)

Reviewer 3 Report

Reviewed article concerns development of an inner finishing method for brass cone pipe via a movable manual electromagnet in a magnetic abrasive finishing process and is write in accordance with generally accepted standards of the scientific works. After careful reading of the submitted text there are some substantive remarks that should be taken into consideration by the Authors to improve reviewed text.

  1. Text of scientific rapports should be writing impersonal.
  2. The abstract should include information about new methods, results, concepts, and conclusions – in its current form, the abstract needs to be rewritten to include more information on novelty of method described in the manuscript.
  3. Literature review should be improved providing more references to recent works from the area of described study.
  1. At the end of the introduction should be clearly and concise given the research gap to create the appropriate lead up for the motivation of the work as well as the aim and novelty of given approach.
  1. I suggest providing more precise information about used measurement systems.
  2. Presented study widely covers defined scientific problem and with experimental investigations provides proper background for given conclusions, however deeper scientific consideration of obtained results referred to the basic phenomena in finishing processes should be given.
  3. The strengths and limitations of the obtained results and applied methods should be clearly described.
  4. The conclusion should be improved in term of the new knowledge gained during analysis, which should be concise with the journal scope.

Author Response

(The authors gave the same response as above.)

Round 2

Reviewer 2 Report

I suggest that the author's revised article should be accepted for publication.

Author Response

Reviewer 2

Comment

I suggest that the author's revised article should be accepted for publication.

Response

Thank you for your positive comments.

Reviewer 3 Report

All my comments were taken into consideration.

Article can be published in its current form.

Author Response

Reviewer 3

Comment: All my comments were taken into consideration. The article can be published in its current form.

Response: Thank you for your positive comments.
